

# A Statistically Optimal Analysis of Systematic Differences between Aeolus HLOS Winds and NOAA's Global Forecast System

Hui Liu[1,2], Kevin Garrett[1], Kayo Ide[3], Ross N. Hoffman[1,2], and Katherine E. Lukens[1,2]

[1]NOAA/NESDIS/Center for Satellite Applications and Research (STAR), College Park, MD

20740, USA

[2]Cooperative Institute for Satellite Earth System Studies (CISESS), University of Maryland,

College Park, MD 20740, USA

[3]University of Maryland, College Park, MD 20740, USA

Correspondence to: Kevin Garrett, NOAA/NESDIS/STAR, 5830 University Research Ct,

College Park, MD 20740, USA. Email: kevin.garrett@noaa.gov. Phone: (301) 683-3641.

Coauthor contact information:
Hui Liu: Hui.Liu@noaa.gov, ORCID 0000-0002-7959-0984.
Kevin Garrett: : kevin.garrett@noaa.gov, ORCID 0000-0002-7444-4363.
Kayo Ide: Kayo.Ide@noaa.gov, ORCID 0000-0001-5789-9326.
Ross N. Hoffman: Ross.N.Hoffman@noaa.gov, ORCID 0000-0002-4962-9438.
Katherine E. Lukens: Katherine.Lukens@noaa.gov.





**Key Points**
• There are speed-dependent systematic differences in the Aeolus M1-bias corrected
Level-2B HLOS winds compared to short-term (6-h) FV3GFS forecasts.
• The total least squares (TLS) regression provides a statistically optimal analysis of the
differences.
• A bias correction based on the TLS bias analysis proposed here is tested in a
companion paper to optimize Aeolus wind assimilation and thus the impact of Aeolus
winds on global NWP forecasts.

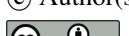



## Abstract

The European Space Agency Aeolus mission launched the first of its kind spaceborne Doppler
wind lidar in August 2018. To optimize assimilation of the Aeolus Level-2B (L2B) Horizontal
Line-of-Sight (HLOS) winds, systematic differences (referred as biases hereafter) between the
observations and numerical weather prediction (NWP) background winds should be removed.
Total least squares (TLS) regression is used to estimate speed-dependent biases between Aeolus
HLOS winds (L2B10) and the National Oceanic and Atmospheric Administration (NOAA)
Finite-Volume Cubed-Sphere Global Forecast System (FV3GFS) 6-h forecast winds. Unlike
ordinary least squares regression, TLS regression optimally accounts for random errors in both
predictors and predictands. Large well-defined, speed-dependent biases are found particularly in
the lower stratosphere and troposphere of the tropics and Southern Hemisphere. These large
biases should be corrected to increase the forecast impact of Aeolus data assimilated into global
NWP systems.

**Key words:** Aeolus winds, Doppler wind lidar, total least squares, bias correction



## 1   Introduction

The space-borne Doppler wind lidar on board the European Space Agency (ESA) Aeolus

mission measures both Rayleigh (i.e., molecular) and Mie (i.e., clouds and aerosols) backscatter

to derive wind profiles along the sensor's Horizontal Line of Sight (HLOS) throughout the

troposphere and lower stratosphere [Straume-Lindner, 2018, Straume et al., 2020]. The Aeolus

HLOS Level-2B (L2B) winds have demonstrated positive impacts on global weather forecasts

[Rennie et al., 2021; Cress, 2020; Garrett et al., 2020, 2021].

To optimize the positive impact of Aeolus winds on weather forecasts, large systematic

differences (referred to as biases hereafter) between Aeolus winds and numerical weather

prediction (NWP) model background winds should be corrected [Daley, 1991]. Therefore, it is

important to identify potential biases between Aeolus winds and their NWP model background

counterparts [Liu et al., 2020, 2021]. The biases may come from both NWP models and Aeolus

winds. First, current operational global NWP models still have larger errors or uncertainty in

regions where conventional observations are sparse or absent, and these errors include bias

components as the NWP models evolve towards their own climatology in the absence of

observations. For example, the backgrounds from the ECMWF model

(https://www.ecmwf.int/en/forecasts) and the NOAA Finite-Volume Cubed-Sphere Global

Forecast System (FV3GFS) model (https://www.gfdl.noaa.gov/fv3/) show large systematic

differences in the zonal winds in the troposphere and lower stratosphere of the tropics, the

Southern Hemisphere (SH), and north of 70° N, with maxima on the order of 2.0, -0.5, and 0.5

m/s, respectively (see Figure 1).



Secondly, although corrections to several substantial bias sources in the Aeolus L2B
winds have been implemented, including corrections to the dark current signal anomalies of
single pixels (so-called hot pixels) on the Accumulation-Charge-Coupled Devices (ACCDs),
linear drift in the illumination of the Rayleigh/Mie spectrometers, and the telescope M1 mirror
temperature variations [Reitebuch et al., 2020; Weiler et al., 2021], uncorrected biases due to
potential calibration issues might remain in Aeolus L2B winds and may contribute to potential
biases between Aeolus and the NWP background winds. The residual biases may lead to sub-
optimal assimilation of Aeolus winds in NWP systems. In addition, the Aeolus L2B winds might
be biased towards the ECMWF model, as the M1 bias correction makes use of ECMWF 6-hour
forecasts [Rennie et al., 2021], which might also lead to sub-optimal assimilation of Aeolus
winds in other NWP systems.
Using ordinary least squares (OLS) to identify and estimate the speed-dependent biases in
the innovations of Aeolus minus NWP background winds (O-B) is subject to contamination from
random errors in Aeolus and/or NWP background winds [Frost and Thompson, 2000], since
OLS assumes no errors in the predictor or independent variable, which in this case would be
either the Aeolus winds, the NWP background winds, or a combination of the two.  In contrast,
total least squares (TLS) regression takes account of errors in both dependent and independent
variables and generates a statistically optimal analysis of the biases [Deming, 1943; Ripley and
Thompson, 1987; Markovsky and Van Huffel, 2007]. For the case of Aeolus and NWP
background winds, the use of linear TLS regression [Ripley and Thompson, 1987] finds a best fit
line that is an estimate of the true (assumed linear) relationship between Aeolus and NWP
background winds.



In this study, the TLS regression approach is used to identify and estimate potential
biases between the Aeolus HLOS winds (L2B10) and the NOAA FV3GFS background winds.
The suboptimality of OLS bias estimates is demonstrated by comparison to the TLS bias
estimates.  A bias correction based on the TLS bias analysis is proposed for the innovations of
Aeolus minus FV3GFS winds in order to optimize Aeolus wind assimilation with the FV3GFS
model and thus improve the impact of Aeolus winds on FV3GFS forecasts.
Section 0 describes the Aeolus L2B and FV3GFS background winds, the TLS bias
analysis method, and the estimation of the ratio of error variances of Aeolus winds to FV3GFS
background winds used in the TLS regression. Section 3 describes the variations of the TLS bias
estimates with height, latitude, and wind speed. Section 4 demonstrates the substantial
differences between the TLS and OLS bias estimates. Section 5 proposes a TLS bias correction
for the O-B innovations. Finally, Section 6 presents a summary of findings and conclusions.
Throughout this article, we will refer to the Aeolus and FV3GFS HLOS winds as the Aeolus and
FV3GFS winds, respectively. In discussions of winds that are not HLOS winds we will use terms
like *u*-wind, *v*-wind, or wind vector.
**2   Data and Methodology**
**2.1   Aeolus L2B and FV3GFS background wind data**
The Aeolus L2B clear-sky Rayleigh winds and cloudy-sky Mie winds are examined for
the period 1-7 September 2019. This one-week period provides a sufficient sample to estimate
the biases. The Aeolus winds were obtained from the Aeolus dataset (L2B10) re-processed by
ESA [Rennie et al., 2021, Weiler et al., 2021]. The reprocessing includes the M1 bias correction,



which removes most of the globally and vertically averaged biases of Rayleigh and Mie winds
[Weiler et al., 2021]. The Aeolus winds are reported at a standard set of vertical layers [de Kloe,
2019, 2020]. This study examines Aeolus Mie and Rayleigh winds within height ranges of 0-16
km and 3-22 km, respectively. These height ranges include almost all Aeolus wind observations.
The height is defined relative to the EGM96 geoid for the L2B winds [Tan et al. 2008].
Similar Aeolus data quality control procedures as recommended by ESA and ECMWF
[Rennie et al., 2021] were implemented to reject the following observations: HLOS L2B
confidence flag "invalid"; Rayleigh winds at layers below 850 hPa, L2B uncertainties greater
than 12 m/s, accumulation lengths less than 60 km, and atmospheric pressure within 20 hPa of
topographic surface pressure; Mie winds with L2B uncertainties greater than 5 m/s and
accumulation lengths less than 5 km.
The winds from Aeolus and collocated FV3GFS backgrounds are obtained from a data
assimilation experiment (hereafter the BASE experiment) where the Aeolus winds are monitored
and the Aeolus wind observation operator ($H_i$) is applied to the FV3GFS background ($\mathbf{x}^b$) to
obtain the value of FV3GFS background wind ($y_i^b = H_i(\mathbf{x}^b)$) corresponding to each Aeolus
observation ($y_i^o$). This experiment employs the FV3GFS data assimilation system, called Global
Statistical Interpolation [GSI, Kleist et al. 2009], configured for the 4DEnVar algorithm with 64
vertical levels, and horizontal resolutions of C384 (~25 km) for the deterministic analysis and
forecast and C192 (~50 km) for the 80 ensemble members [Wang and Lei, 2014].
When examining Aeolus wind statistics, we stratify the Aeolus data by orbital phase,
either ascending when the spacecraft is moving northward or descending when the spacecraft is
moving southward. The vertical and daily variations of global horizontal means and standard



deviations of the innovations of Mie winds minus FV3GFS background winds are consistent
throughout the period (Figs. 2 and 3). For Mie winds in ascending orbits, the biases are positive
above 6 km and negative below 6 km, as large as +1.8 m/s and -0.5 m/s, respectively. The biases
are smaller and positive at most levels in the descending orbits. The standard deviations are
smallest (about 4 m/s) from 2 to 8 km elevation and increase to only about 5 m/s at the highest
levels. For Rayleigh winds in descending orbits, the biases are as positive as +1.2 m/s above 10
km, and as negative as -1.2 m/s below 8 km. The positive biases in ascending orbits are smaller.
The standard deviations are smallest (again about 4 m/s) from 6 to 12 km elevation and increase
to about 7 m/s at the highest levels. The results indicate that the biases vary substantially with
height for both Mie and Rayleigh winds, the standard deviations vary from 4 m/s to somewhat
larger values at higher elevations, and that both mean and standard deviations remain stable in
time throughout the period.

The mean differences of Mie and Rayleigh winds minus FV3GFS winds vary

considerably with latitude (Figure 4). Mie winds have biases as large as +1.5 m/s in the upper
troposphere and Rayleigh winds have biases as large as +2.0 m/s in the tropical upper
troposphere. Both Mie and Rayleigh winds show negative biases as large as -1.0 m/s in the
lowest layers.
**2.2    TLS Linear Regression**

In this section, we review the TLS regression method [Ripley and Thompson, 1987] in

the context of estimating potential speed-dependent biases between Aeolus winds and FV3GFS
background winds. The TLS estimate for each collocated pair of Aeolus and FV3GFS winds ($y_i^o$,
$y_i^b$) is defined by





$$y_i^o = \hat{y}_i^o + \varepsilon_i^o \quad \text{and} \quad y_i^b = \hat{y}_i^b + \varepsilon_i^b \qquad (i = 1, N) \qquad (1)$$
where $\hat{y}_i^o$ and $\hat{y}_i^b$ are the TLS estimates of the true Aeolus and FV3GFS winds, and $\varepsilon_i^o$ and $\varepsilon_i^b$
are random errors, and N is the number of Aeolus/FV3GFS collocations in the sample. The
sample might be defined by a vertical layer or a latitude band. In OLS regression, since it is
assumed that there are no errors in the predictor, the predictor can be used directly to estimate the
predictand. The situation is a little more complicated in TLS regression where $(\hat{y}_i^b, \hat{y}_i^o)$, the most
probable true state, is the point on the regression line that is closest in a statistical sense to the
point $(y_i^b, y_i^o)$.
Here we assume that $\varepsilon_i^o$ and $\varepsilon_i^b$ are independent and that the random error variance ratio
$\delta = (\sigma^o/\sigma^b)^2 = \text{E}[\varepsilon_i^o \varepsilon_i^o] / \text{E}[\varepsilon_i^b \varepsilon_i^b]$ is known. Also, we assume the true relationship between the
Aeolus and FV3GFS winds is described by a linear function:
$$\hat{y}_i^o = c_0 + c_1 \hat{y}_i^b \qquad (i = 1, N) \qquad (2)$$
where $c_0$ is an offset or constant bias and $c_1$ is a speed-dependent bias coefficient.
The TLS regression finds an optimal estimate of the $\hat{y}_i^b$, $c_0$ and $c_1$ by minimizing the
cost function
$$J = \sum_{i=1}^{N} \left( (\varepsilon_i^o/\sigma^o)^2 + \left(\varepsilon_i^b/\sigma^b\right)^2 \right)$$
$$= \frac{1}{(\sigma^o)^2} \sum_{i=1}^{N} \left( \left(y_i^o - c_0 - c_1\hat{y}_i^b\right)^2 + \delta\left(y_i^b - \hat{y}_i^b\right)^2 \right)$$
To determine the $\hat{y}_i^b$, set the derivative of J with respect to $\hat{y}_i^b$ to zero, to obtain
$$\hat{y}_i^b = (c_1(y_i^o - c_0) + \delta y_i^b) / (c_1^2 + \delta) \qquad (i = 1, N) \qquad (3)$$





Eq. (3) thereby reduces the problem to a minimization in terms of $c_0$ and $c_1$. A similar equation
holds even if the error variances vary with $i$, but then there is no closed form solution for $c_0$ and
$c_1$, as there is in the current case, which is known as the Deming problem [Ripley and
Thompson, 1987]. When the coefficients $c_0$ and $c_1$ are obtained, the TLS estimate for the new or
within-sample observation is given by Eq. (3).  Finally, the estimate of the bias for the $k$th
observation, either for a new or within-sample observation, is given by
$$\hat{d}_k = \hat{y}_k^o - \hat{y}_k^b = c_0 + (c_1 - 1)\hat{y}_k^b \tag{4}$$

We will refer to $c_0$ and $(c_1 - 1)$ as the constant and speed-dependent bias coefficients,
respectively, hereafter.

Note that the error variance ratio $\delta$ is a crucial parameter in the TLS bias analysis. If

$\sigma^o = 0$ or $\sigma^b = 0$, then the TLS solution is equivalent to the OLS regression of the O-B on the
Aeolus winds or on the FV3GFS winds, respectively.
**2.3    Estimation of the random error variance ratio**

The random error variance ratio $\delta = (\sigma^o / \sigma^b)^2$ used in the TLS bias analysis is estimated

from the O-B innovations from the BASE experiment using the Hollingsworth-Lonnberg (HL)
method [Hollingsworth and Lonnberg, 1986]. It is assumed that there is no correlation between
the random errors in Aeolus and FV3GFS winds and no horizontal correlation in the random
errors in Aeolus winds at 90 km distance and beyond. For more details, see Hollingsworth and
Lonnberg [1986] and Garrett et al. [2021].

The random error variance ratio δ is estimated at the middle height of each vertical range

bin using the Aeolus samples for 1-7 September 2019, separately for Mie and Rayleigh winds.



Figure 5 shows that the vertical profiles of the square root of δ varies in the range of 1.2-1.6 and
2-3 for Mie winds versus FV3GFS winds and Rayleigh winds versus FV3GFS winds,
respectively.

## 3     The TLS Bias Estimates

The statistical relationship between Aeolus and FV3GFS winds is illustrated by the

density plots of collocated Aeolus and FV3GFS winds in a single layer shown in Figure 6. There
is a strong correlation of 0.93 between Mie and FV3GFS winds, and of 0.94 between Rayleigh
and FV3GFS winds. The TLS analyses of the FV3GFS winds versus Aeolus winds indicate that
the innovations (Aeolus minus FV3GFS winds) are positive and increase with wind speed. In
terms of Eq. (4), for Figure 6a, the innovation solution is 0.53 m/s + 0.06 times the background
solution, while for Figure 6b, the innovation solution is 1.04 m/s + 0.04 times the background
solution.

### 3.1     Variation of Biases with Height

The variation of the TLS solution with height and orbital phase is described here. The

TLS samples are over all latitudes. The vertical distribution of the TLS constant and speed-
dependent bias analysis coefficients for the innovation in terms of the background in Eq. (4) is
shown in Figure 7. The speed-dependent bias coefficient $(c_1 - 1)$ varies substantially with height
and orbital phase. For Mie winds, the coefficient is quite large at most heights, ranging from 3 to
6%, with maxima at 3 km and 12-16 km. The coefficient for Rayleigh winds is smaller and
ranges from 1 to 3% in ascending orbits and 1 to 5% in descending orbits, with maxima around
the 3.5 and 16 km.





The constant bias coefficient $c_0$ for both Mie and Rayleigh winds also shows large
variations on height and orbit with its value as large as +/- 1.0 m/s. In general, the constant bias
coefficient is positive in upper layers and negative in layers close to the Earth surface, consistent
with the patterns seen in the global horizontal average of innovations in Figures 2 and 3.
The vertical distribution of the average TLS bias estimates as function of Aeolus wind is
shown in Figure 8. The average TLS biases vary substantially with height. Since the TLS biases
are in part dependent on speed, at most heights the biases increase substantially as the magnitude
of Aeolus wind speed increases. The biases at high Aeolus wind speeds are considerably larger
for Mie winds than for Rayleigh winds, as large as +2.5 m/s and -2.0 m/s for Mie winds, and
+1.5 m/s and -2.0 m/s for Rayleigh winds. There are clear speed-dependent biases in the vertical
average of these biases (Figure 9). The results suggest that both vertically varying and vertically
averaged speed-dependent biases remain in the Aeolus winds (L2B10).
**3.2   Variation of Biases with Latitude**
The variation of the TLS solution with latitude and orbital phase is described here. The
TLS samples are over all heights for 10-degree latitude bands. In general, the coefficients
obtained are large and vary considerably with latitude and orbital phase, with maxima found in
the tropics (Figure 10). For example, the speed-dependent bias coefficient $(c_1 - 1)$ for Mie
winds in the tropics can be quite large, ranging from 0% to a maximum of 11%. The coefficient
$(c_1 - 1)$ is smaller for Rayleigh winds, ranging from -1% to 5%, with maxima found in the
tropics and at northern high latitudes. The constant bias coefficient $c_0$ for Mie winds also varies
considerably with latitude and orbit, ranging from -1.0 m/s to +1.6 m/s. The coefficient $c_0$ is
smaller for Rayleigh winds.



The latitudinal distribution of the average TLS bias as a function of Aeolus wind is
shown in Figure 11. For Mie winds, the average TLS bias increases considerably at most
latitudes as the magnitude of Aeolus wind speed increases, particularly in the tropics and SH,
with maxima of about +/-2.5 m/s. For Rayleigh winds, the average biases are much smaller and
are consistent with the fact that the M1 bias correction removes most globally and vertically
averaged biases of Rayleigh winds [Weiler et al., 2021].
**3.3     Discussion**
The results presented in this section indicate that the speed-dependent bias coefficient is
quite large, with $(c_1 - 1)$ reaching up to ~10% and 5% for Mie and Rayleigh winds,
respectively, particularly in the lower stratosphere and lower troposphere of the tropics. This
suggests that there exist large speed-dependent biases in FV3GFS background winds and/or in
the Aeolus winds. Given that there exist large uncertainties in the FV3GFS (and ECMWF)
background winds in the tropics (see Figure 1), it is likely that the FV3GFS may be a significant
source of the large biases and this will require further investigation. In any case, these large
speed-dependent biases should be corrected to optimize Aeolus wind assimilation and the impact
of Aeolus winds on NWP forecasts.
**4     Comparison to OLS Regressions**
As a comparison to the TLS bias estimate results, we conducted parallel OLS regressions
using three different predictors of the biases in O-B. These predictors are the FV3GFS winds, the
Aeolus winds, and their average. The first two of these OLS regressions are equivalent to OLS
regressing Aeolus on FV3GFS winds and OLS regressing FV3GFS on Aeolus winds. As





examples, the regression lines of these two cases are added to Figure 6. The TLS speed-
dependent coefficient $(c_1 - 1)$ (in Eq. 4) = 6% and 4% for Mie and Rayleigh winds,
respectively. However, the OLS regression of Aeolus winds on FV3GFS winds produces
considerably smaller bias estimates, with $(c_1 - 1)$ estimated as 1% and 2% for Mie and
Rayleigh winds, respectively; thus, this OLS regression considerably underestimates the biases.

On the other hand, the OLS regression of the FV3GFS winds on Aeolus winds exhibits

much larger bias estimates relative to the TLS bias analysis, with $(c_1 - 1)$ estimated as 18% and
15% for Mie and Rayleigh winds, respectively. This indicates that the speed-dependent biases
are considerably overestimated by the OLS regression on Aeolus winds.

The vertical distributions of the average biases as a function of Aeolus winds are shown

in Figure 12 for the descending orbits for three methods: The top panels are for OLS regression
using FV3GFS winds as a predictor, the middle panels, which repeat the bottom two panels of
Figure 8 are for TLS regression, and the bottom panels are for OLS regression using the average
of FV3GFS and Aeolus as a predictor (bottom). The average bias estimates in the top panels are
about 0.5-1.0 m/s smaller in magnitude in most layers than the middle panels. This confirms that,
on average, the biases are considerably underestimated by OLS regression using FV3GFS winds
as a predictor.

The average biases in the bottom panel are about 0.5-1.5 m/s in magnitude larger than the

middle panel in most layers, particularly for Rayleigh winds, indicating the biases are
overestimated by OLS regression using the average of Aeolus and FV3GFS winds as a predictor.
The bias estimates of OLS regression using Aeolus winds only as a predictor (not shown) are
even larger (than the bottom panel).





## 5    A TLS Bias Correction


In this section, a TLS bias correction for O-B is proposed to optimize Aeolus wind data

assimilation. For each assimilation cycle, the bias coefficients are computed by TLS regression
for the O-B in the week before the cycle (i.e., for the previous 28 cycles). One week provides a
large enough sample for the regression. As shown by Ripley and Thompson [1987], the TLS
solution only involves solving a quadratic equation with coefficients given by sample sums.
Therefore, an efficient approach is to calculate and save these sums for every cycle and
accumulate them over the 28 cycles. Because the findings in this study show substantial variation
of the bias coefficients with latitude, vertical layer, and orbital phase, the bias coefficients are
calculated from the winds in 19 discrete bins of latitude (centered every 10º between 90° S to 90°
N) for each vertical range/layer and for ascending and descending orbits separately. For each of
the O-B innovations in the assimilation cycle, values of $c_0$ and $c_1$ are linearly interpolated to the
latitude of the Aeolus observation. Subsequently, the TLS estimated bias, calculated using Eq.
(4), is subtracted from the O-B. Note that the bias correction is determined by the TLS analysis
solution for $\hat{y}_k^b$ that in turn is determined from the observation and background wind, $y_k^o$ and $y_k^b$,
following Eq. (3).

The proposed scheme is applied to the O-B innovations of the BASE experiment. The

vertical distribution of the average remaining biases as a function of Aeolus wind is shown in
Figure 13, which is in the same format and for the same sample of observations as Figure 8. A
comparison of these two figures reveals that most of the biases are removed by the proposed TLS
bias correction. The latitudinal variations of the biases are also corrected (Figure 14). In addition,
the biases in the vertical average are also mostly removed, as shown in Figure 9.



## 6   Summary and Conclusions


In this study a TLS regression is used to optimally estimate speed-dependent biases
between Aeolus L2B Horizontal Line-of-Sight winds and short-term (6-h) forecasts of NOAA's
FV3GFS. The winds for 1-7 September 2019 are analyzed. Clear speed-dependent biases for
both Mie and Rayleigh winds are found, particularly in the lower troposphere and stratosphere of
the tropics and Southern Hemisphere. The largest biases are about 10% and 5% of FV3GFS wind
speed, as large as +/- 2.5 m/s and +/- 1.5 m/s at high FV3GFS wind speed, for Mie and Rayleigh
winds, respectively.
It is found that the biases are considerably underestimated by the OLS regression of the
innovations of Aeolus winds minus FV3GFS background winds on FV3GFS winds; but are
overestimated by the OLS regression, both on Aeolus winds only, and on the average of Aeolus
and FV3GFS winds.
The biases should be fully corrected to optimize Aeolus wind assimilation and to improve
the impact of Aeolus winds on FV3GFS global forecasts. The proposed TLS bias correction can
remove most of the biases before assimilation. In a companion paper, Garrett et al. [2021]
demonstrate that the application of this TLS bias correction to the Aeolus minus FV3GFS
background (O-B) winds considerably enhances the positive impact of Aeolus winds on NOAA
FV3GFS global and tropical cyclone forecasts. It is expected that the application of this
additional bias correction to the O-B innovations of Aeolus winds can improve and enhance
Aeolus data impacts on the analysis and forecast skill of other NWP systems as well.
Note that the proposed TLS approach presented here might be applied to other types of
observations that have errors typically characterized as a percentage of the observed value,



including quantities related to the concentrations or mass fractions of chemical species or
hydrometeors, or quantities like radio occultation refractivity and bending angle.

**Acknowledgments**

This work was supported by the NOAA/NESDIS Office of Projects, Planning, and

Acquisition (OPPA) Technology Maturation Program (TMP), managed by Patricia Weir and Dr.
Nai-Yu Wang, through the Cooperative Institute for Satellites and Earth System Studies
(CISESS) at the University of Maryland (Grant NA14NES4320003 and NA19NES4320002).
The authors would like to acknowledge Dr. Michael Rennie (ECMWF) and Dr. Lars Isaksen
(KNMI) for their comments and suggestions on the assimilation of Aeolus observations, and Dr.
William McCarty with NASA/GMAO for providing earlier versions of the GSI with Aeolus
ingest and observation operator capability. The Aeolus L2B BUFR data were provided by
ECMWF. The scientific results and conclusions, as well as any views or opinions expressed
herein, are those of the author(s) and do not necessarily reflect those of NOAA or the U.S.
Department of Commerce.

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





**8 Figures**

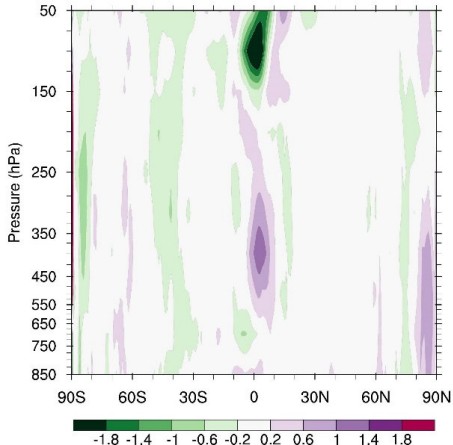


Figure 1. Latitudinal and height distributions of zonal mean difference of ECMWF minus FV3GFS
background zonal wind (m/s) for 1-7 September 2019.





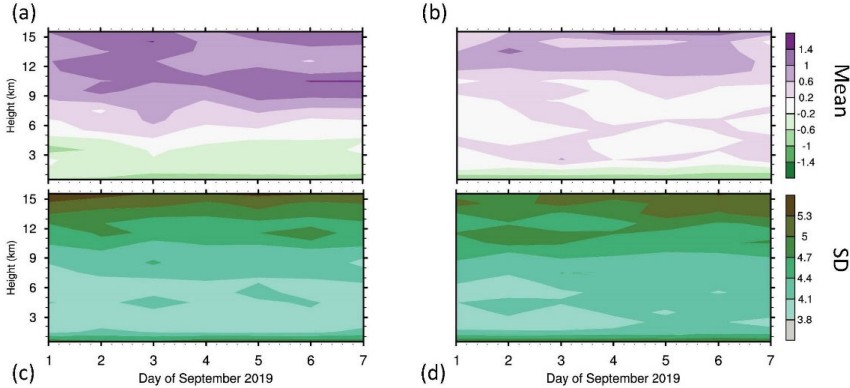


Figure 2. Vertical and daily variations of global horizontal means (a, b) and standard deviations (c, d) of

the innovations of Mie winds minus FV3GFS background winds (m/s) in ascending (a, c) and descending

(b, d) orbits.


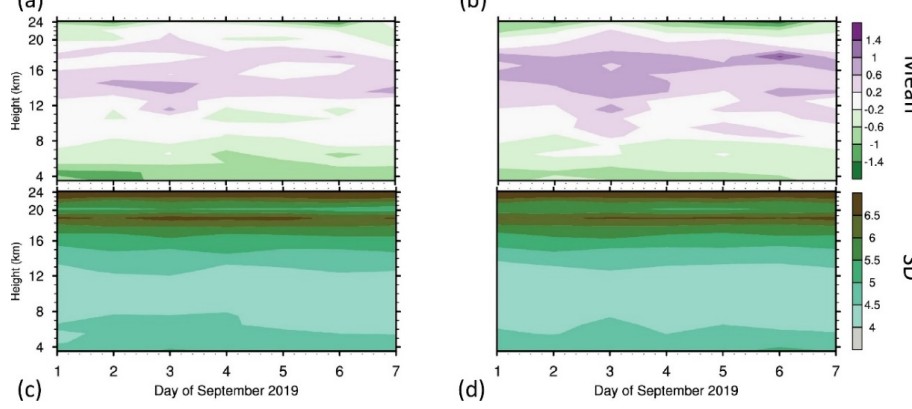

Figure 3. As in Fig. 2 but for Rayleigh winds.



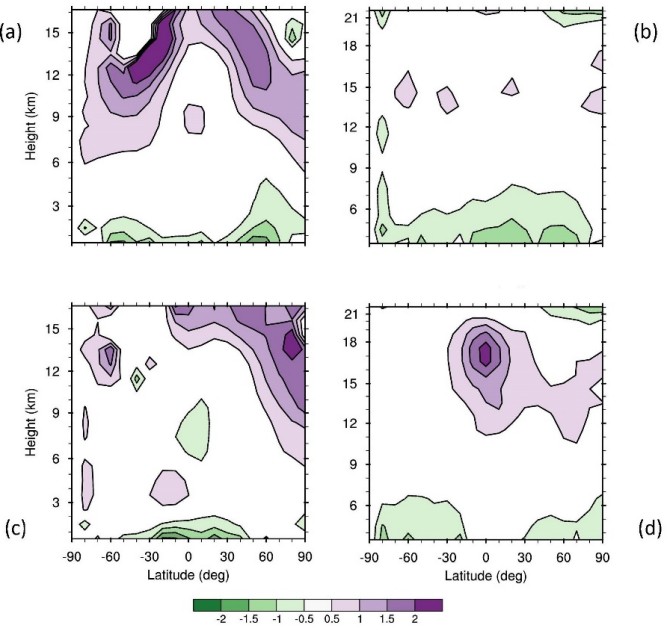


Figure 4. Latitudinal and height distributions of mean differences (color scale, m/s) of Mie minus

FV3GFS winds (a, c) and Rayleigh minus FV3GFS winds (b, d) in ascending (a, b) and

descending (c, d) orbits for 1-7 September 2019.




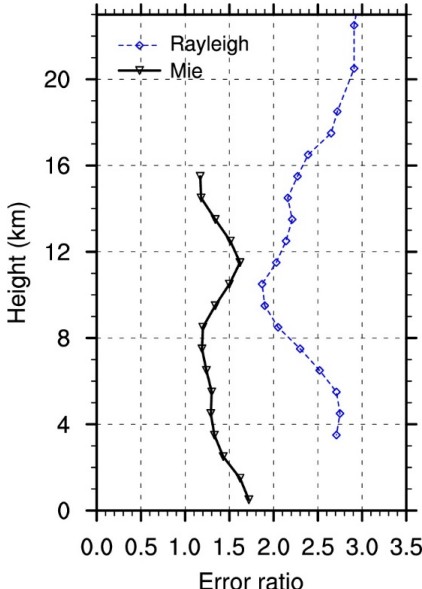


Figure 5. Vertical variation of the square root of the ratio of random error variance in Aeolus winds
versus FV3GFS background winds for Mie (solid black) and Rayleigh (dashed blue) winds. Results are
based on global O-B innovations of Aeolus minus FV3GFS winds from the Aeolus BASE experiment
using Hollingsworth-Lonnberg method. The symbols are plotted at averaged height in each vertical layer.





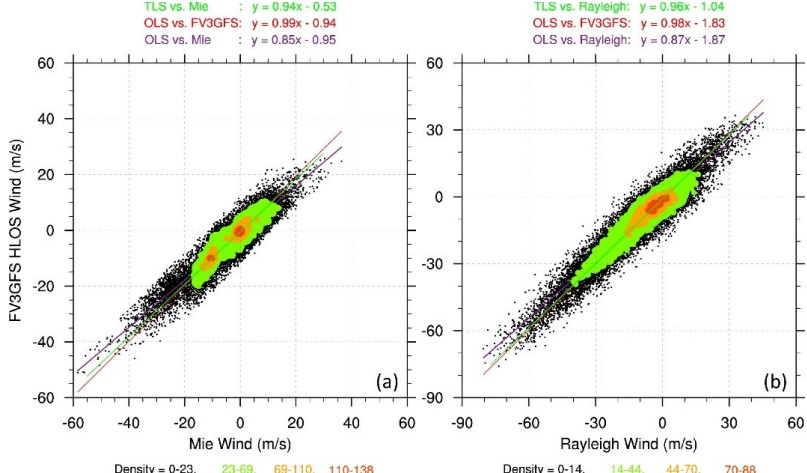

Figure 6. Density plots of global collocated Mie and FV3GFS winds at ~3.5 km altitude (a), and Rayleigh

and FV3GFS winds at ~16.5 km altitude (b) in descending orbits. The TLS analysis line (green), the OLS

regression line of FV3GFS winds on Aeolus winds (purple), and the OLS regression line of Aeolus winds

on FV3GFS winds (transformed and plotted as a function of Aeolus winds in red) are shown, with

corresponding regression coefficients displayed above each panel.





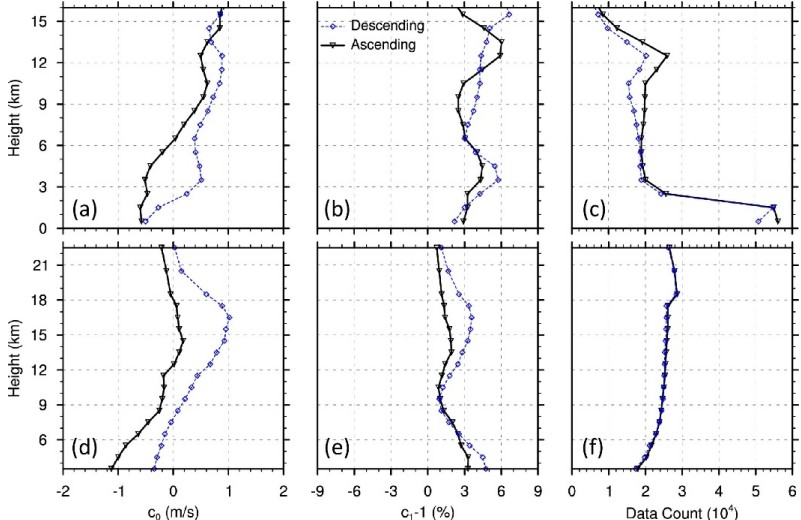

Figure 7. Vertical variations of TLS bias coefficients for Mie versus FV3GFS winds (a, b, c), and

Rayleigh versus FV3GFS winds (d, e, f). Each point plotted represents a separate TLS analysis for all

observations in each layer for all latitudes and for either ascending (black) or descending (blue) orbits.

The symbols are plotted at the average height of the observations in each layer.

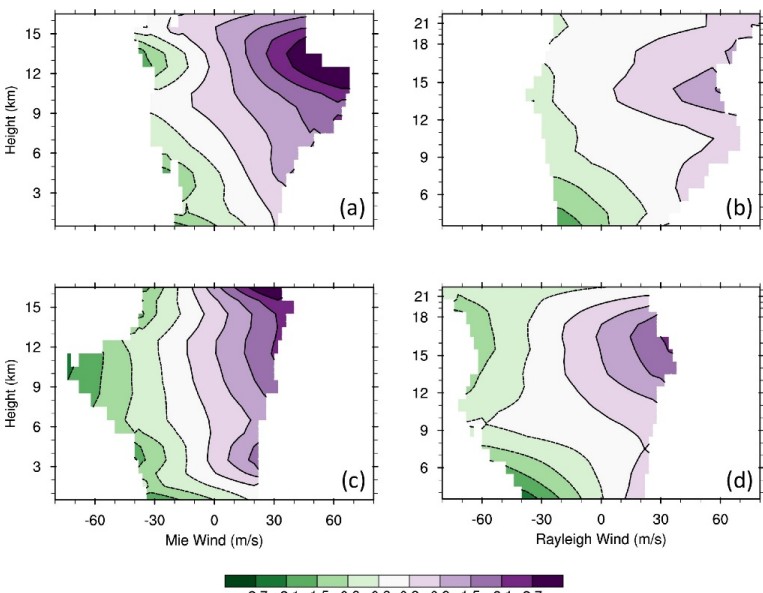

Figure 8. Vertical distributions of average TLS estimated biases (color scale, m/s) for Mie versus

FV3GFS winds (a, c) and Rayleigh versus FV3GFS winds (b, d) as a function of observed Aeolus winds

(m/s) in ascending (a, b) and descending (c, d) orbits for all latitudes, obtained from the TLS fits

displayed in Figure 7.

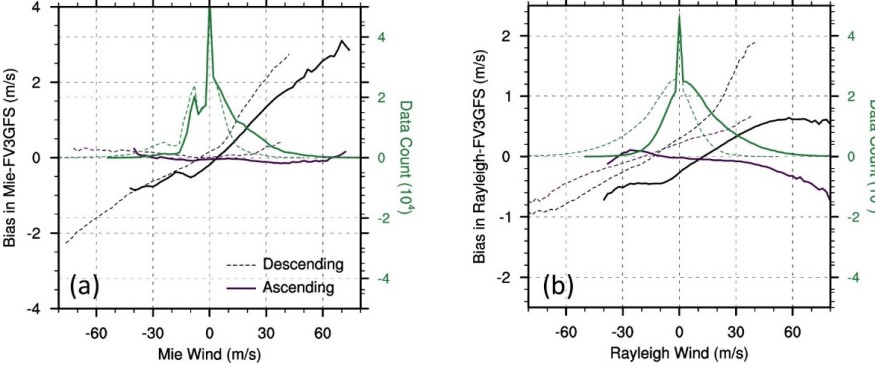

Figure 9. TLS estimated biases (m/s) before (black lines) and after (purple lines) TLS bias correction for

Mie versus FV3GFS winds (a) and Rayleigh versus FV3GFS winds (b) as a function of the observed



Aeolus winds (m/s), vertically averaged for all latitudes of Aeolus winds. The green lines report the
number of observations in each 2 m/s bin.

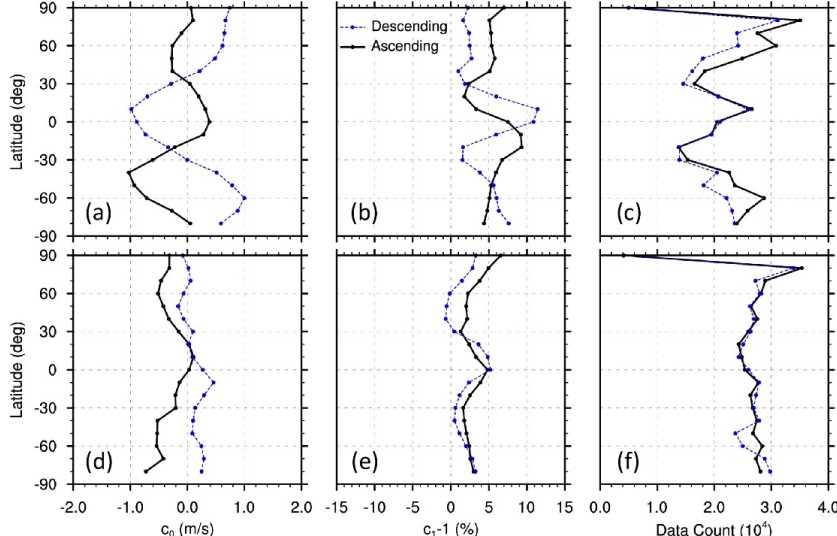

Figure 10. Latitudinal variation of TLS bias coefficients for Mie versus FV3GFS winds (a, b, c) and for
Rayleigh versus FV3GFS winds (d, e, f). Each point plotted represents a separate TLS analysis for all
observations in all vertical layers in a 10° latitude band for either ascending (black) or descending (blue)
orbits. The latitude bands are centered every 10° from 90°S to 90°N. The symbols are plotted at the center
in each latitude band. The vertical layers are 0-16 km for Mie winds and 3-22 km for Rayleigh winds.




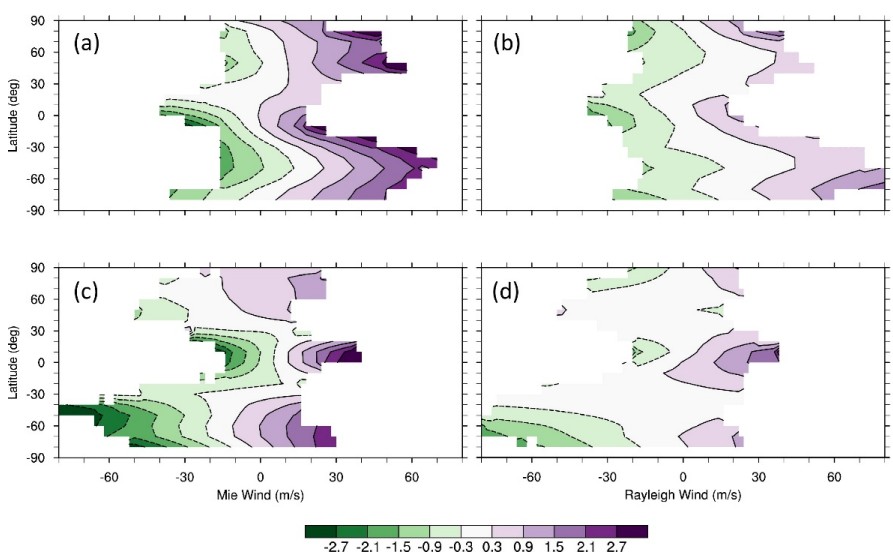


Figure 11. Latitudinal distributions of average TLS estimated biases (color scale, m/s) for Mie versus
FV3GFS winds (a, c) and Rayleigh versus FV3GFS winds (b, d) as a function of Aeolus wind in
ascending (a, b) and descending (c, d) orbits for all latitudes, obtained from the TLS fits displayed in
Figure 10.





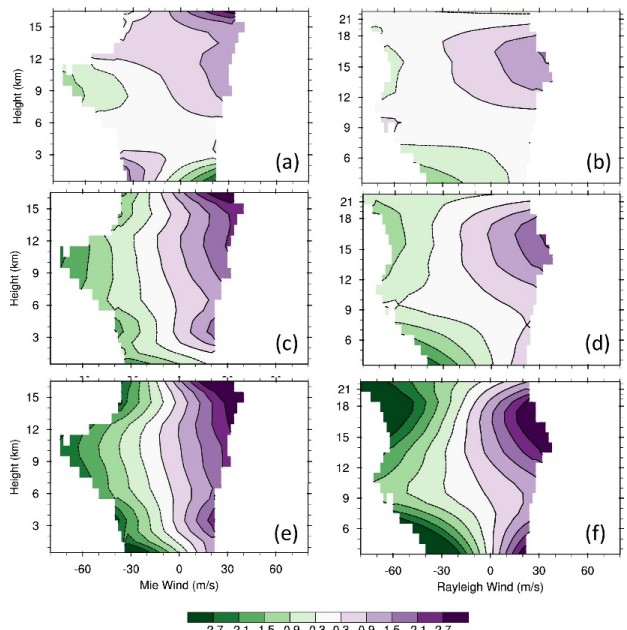

Figure 12. Vertical distributions of average bias estimates (color scale, m/s) in Mie versus FV3GFS winds

(a, c, e) and Rayleigh versus FV3GFS winds (b, d, f) as a function of Aeolus winds using one of three

methods for descending orbits for all latitudes. The methods are OLS using FV3GFS winds as a predictor

(a, b), TLS (c, d, same as the bottom panels of Figure 8), and OLS using the average of Aeolus and

FV3GFS as a predictor (e, f).



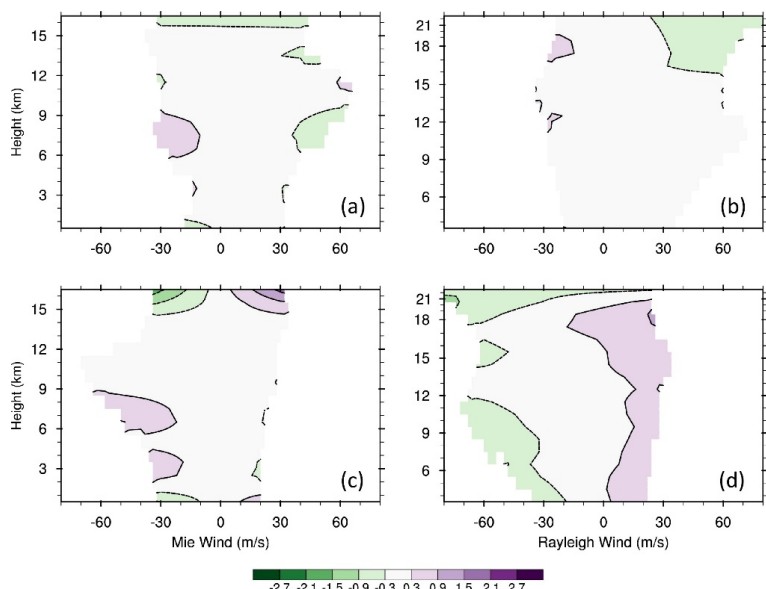


Figure 13. As in Figure 8 but for the mean innovation (O-B) after the TLS bias correction is applied. For

each 6-h cycle during 1-7 September 2019, the TLS bias correction is calculated from the 28 preceding

cycles.





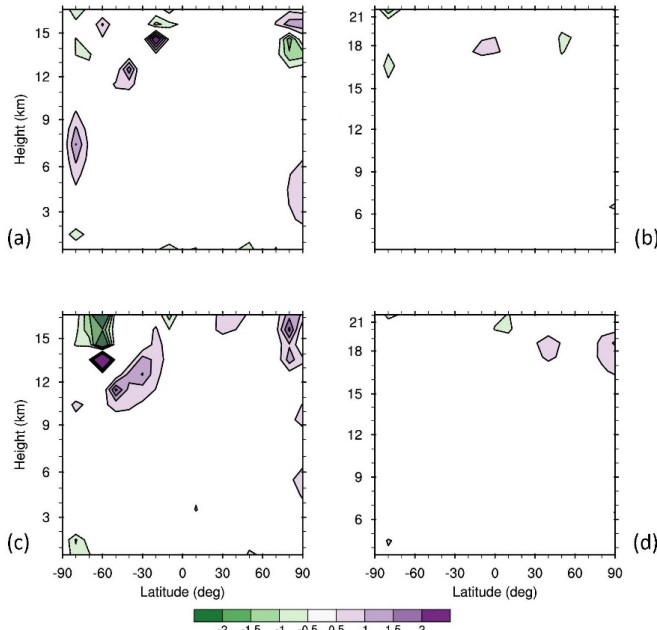

Figure 14. As in Figure 4 but after the TLS bias correction is applied. Note that the remaining
bias in several bins are due to small sample size, and the TLS bias correction is not applied in
these bins in Aeolus wind assimilation.