# Peer review of "A Statistically Optimal Analysis of Systematic Differences between Aeolus HLOS Winds and NOAA's Global Forecast System"

_Atmospheric Measurement Techniques, 2022_

## Referee Comment (RC2)

**Review of the paper „A statistically optimal Analysis of systematic differences between Aeolus HLOS winds an NOAA's Global Forecast model"**

The paper describes a bias correction method, based on the on observation minus background statistics and relative error variances between the model background and Aeolus wind data. The paper showed that the first reprocessed HLOS winds (B10) used in this study still have large remaining mean differences compared to the FV3GFS model background. These biases vary with latitude, height, orbit phase and HLOS wind speed. To correct these biases a total least square (TLS) regression method was used. Here the authors assumed a linear dependency of the bias with wind speed. The TLS method was calculate for each orbital phase, vertical layer and 10-degree latitude band over a period of 7 days. Separated for both, Mie and Raleigh channels. The necessary error variances were estimated using the Hollingsworth-Lonnberg method, which is based on obs minus background statistics. In a comparison to the normal linear regression methods the strength of the TLS method was shown.

In the following, I have only some small suggestions for some minor revisions

Lines 62-70. First, you said that NWP models have large uncertainties in regions where conventional observations are missing. Do you mean analysis uncertainties or forecast uncertainties? Also with observations, the models tend to evolve towards their own climatology over long forecast lead times because the forecast errors become much larger than the analysis errors. Can you clarifiy these sentences a little, and can you you taken into account the different use of satellite radiances between the different centers which can explain a lot of the differences you mentioned in data sparse regions.

It is also not clear to me which backround you use in Fig. 1 and further. In a 4d-VAR or 4D-ENVAR the background is a time series of forecasts over the used assimilation window. Which forecast lead time do you use ?

Line 78-81: I think that the use of ECMWF model fileds to crrect the M1 temperature bias can be neglected. Also to derive other observations like AMVs different model fields are used but their impact can be neglected almost all the time

Line 96: Innovation of Aeolus minus model is a double meaning since innovation means per definition obs minus background. Let the "Innovation of" away

Line 99: You certainly mean Section 2 instead of section 0

Line 119-124: Here you describe the quality control steps recommended by ESA and ECMWF. Do you use additional quality control steps like background check or variational quality control check etc. ?

Line 166 – 182: You assume that the Aeolus bias depends linearly on the Aeolus wind speed. Do you have any prove of this assumption. The bias can also have a non-linear part and if so do you have any idea how large the non-linear part can be.

Chapter 3: To show, that the innovations depend on Aeolus wind speed you can include a picture where on the y-axes you display the innovations and on the x-axes you display the Aeolus wind speed.

Chapter 3.1: In Fig. 9 and including text you describe the success of using the TLS method to reduce the speed dependent bias. Can you explain the high peak of your data count at zero wind speed? It seems the you bias correction enhance the negative bias for large positive wind speeds in case of Rayleigh winds. Can you explain that.

Chapter 3,2; In Figure 11 there are also larger biases visible in the Mie descending orbit for small wind speeds in the tropics. Can you explan that ?

Section 4: In Fig. 12 we see, that the differences between the TLS method and the different OLS regressions are relatively large although the same linear model and assumptions are used. Can you explain this ? You said that using OLS statistics underestimates the biases. How can I see that? What is the truth ?

At the end I think it would be a nice thing to show some assimilation and forecast results using your TLC method in comparison to not using your bias correction. At least some results from your assimilation cycle.

---

## Author Comment (AC2)

**Response to comments of Anonymous Referee #1 (RC1) on AMT preprint "A Statistically Optimal Analysis of Systematic Differences between Aeolus HLOS Winds and NOAA's Global Forecast System" by Hui Liu et al., Atmos. Meas. Tech. Discuss., https://doi.org/10.5194/amt-2022-20**.

This manuscript describes the additional bias correction applied to the first reprocessed HLOS winds (B10) that has been developed for the assimilation of Aeolus data in the NOAA FV3GFS model. It is first shown that there are important remaining mean differences between the B10 HLOS winds (O) and corresponding FV3GFS background values (B), particularly in the tropics. These mean differences (referred to as biases in this study), vary with latitude, height, HLOS wind speed and orbit phase. It is assumed that these biases vary linearly with wind speed. A total least squares (TLS) regression is used to calculate the biases for each orbit phase, vertical layer and 10-degree latitude band over a period of 7 days for both Mie and Rayleigh winds. The TLS regression takes into account of the relative error variance between the FV3GFS background and Aeolus HLOS winds, leading to more optimal bias estimates than those from ordinary least squares regression. The error variances are estimated using the Hollingsworth-Lonnberg method, which is also based on O and B statistics.

Two important aspects are missing in this manuscript in order to make it standalone: the merit of using the Hollingworth-Lonnberg method for estimating the error variance ratio and the impact of the proposed bias correction scheme on forecasts. These two aspects are reported in a companion paper recently submitted to QJRMS by the same authors. However, since the manuscript is relatively short, I suggest either merging it with the companion paper or expanding the manuscript by including more details on the Hollingworth-Lonnberg method and by adding some forecast impact results. There are also a number of places in the manuscript where clarifications are needed, as described in the following comments and suggestions.

*Thanks for your suggestions. More details of the Hollingworth-Lonnberg method have been added in Section 2.3, and new Section 6 describes an Aeolus data impact experiment and forecast results for a record-breaking winter storm over the US.*

Lines 62-65 : '…global NWP models still have larger errors in regions where conventional observations are sparse…' Is this statement based on the difference between analyses or shortrange forecasts from different NWP centers?  Which type of error is most important in these regions?  It is true that '…NWP models evolve towards their own climatology in the absence of observations…'. However, the various geostationary and polar orbiting satellites now provide nearly global coverage of MW and IR radiances in the troposphere and lower stratosphere every 6 hours such that NWP models do not evolve towards their own climatology but towards the analysis largely constrain by the assimilation of satellite radiances and derived products. Please clarify this sentence.

*Here we refer to both mean and random errors in the 6-h forecast of winds in the upper troposphere and lower stratosphere of the Tropics, as well as in the lower troposphere of the Southern Hemisphere. The large errors in winds might be related to the lack of wind observations as well as differences in using other observations, e.g., the satellite radiance data, in these regions.*

*The satellite radiance data do constrain NWP analyses and forecasts.  The text is revised and the sentence of '...NWP models evolve towards their own climatology in the absence of observations…' is removed.*

*The manuscript now reads "Such systematic differences in regions where conventional data are sparse may be due in part to differences in the assimilation of satellite radiances at the NWP centers".*

Lines 67-70 : It is not clear what it is shown in Figure 1. In 4D data assimilation systems, such as 4D-Var and 4D-EnVar, the background is a time series of forecast fields over the assimilation window used to calculate O-B at the 'appropriate' observation time. Which forecast lead-time is shown in Figure 1?

*We agree, this was not clear. As now stated in the Figure 1 caption, this figure shows the differences in the 6-h forecasts initialized from ECMWF and FV3GFS analysis at 00, 06, 12, and 18Z.*

Line 79 : The M1 bias correction scheme makes use of the ECMWF O-B distributed over 12-h assimilation windows, not 6-h forecasts. It is true that the use of ECMWF background in the M1

bias correction scheme may have a detrimental effect on the assimilation of Aeolus winds in other NWP systems. However, this effect is expected to be small since the regression (M1 temperatures vs O-B) is made over the globe, not locally.

*This sentence is removed as suggested by Reviewer #2.*

Lines 83, 96, 104, 136, 295, 300, 324, 419, 432 : The definition of innovation in the meteorological data assimilation field is O-B. Stating 'innovations of observation minus background' is redundant, as well as 'O-B innovations'.

*Thank you. We now say innovations everywhere.*

Line 99 : Change Section 0 by Section 2.

*Corrected.*

Lines 119-124: This paragraph described the quality control applied to the Aeolus winds as recommended by Rennie et al. (2021). It is however not mentioned whether or not an additional quality control (e.g. background check) is applied to remove remaining outliers. Could you elaborate on this?

*The manuscript now states that "Further, a standard outlier check rejects any Aeolus wind for which $|y_i^o - y_i^b|$ is greater than 4 times the estimated errors for Aeolus winds prescribed by the data assimilation system".*

Line 167-170 : It is assumed in this study that the bias vary linearly with background HLOS wind speed. This is not well justified and needs to be further discussed in this manuscript. For instance, recent work by Marseille et al.(2021) (NWP calibration applied to Aeolus Mie channel winds - Marseille - - Quarterly Journal of the Royal Meteorological Society - Wiley Online Library ) has shown that the bias for the Mie winds is complex and varies nonlinearly with wind speed. In this context, how can a linear model for the HLOS wind bias be justified?

*The scatterplot and averages of Aeolus innovations as a function of Aeolus wind speed are shown in the new Figure 5. This figure suggests the existence of both linear and non-linear*

*bias components as a function of Aeolus wind speed. In this study, we focus on the estimation of the linear component of the biases. We plan to address the nonlinear component of the biases using a nonlinear TLS bias correction in a future study. This discussion has been added to the text at the end of Section 2.1.*

Line 187 : It is true that the OLS regression assumes that the predictors (B, O or (B+O)/2 here) are free of errors. However, the OLS regression cannot be formulated using the same TLS cost function and by setting sigma O or sigma B to zero. Instead of elaborating more on this, I suggest removing this statement since the OLS regression is described in Section 4.

*Done.*

Line 190 : Why are the estimated L2B standard deviation errors for the HLOS winds not used for specifying sigma O?

*The manuscript now states "In this study, errors of Aeolus winds are estimated by the Hollingsworth-Lonnberg method (Hollingsworth and Lonnberg, 1986; Garrett et al., 2022), which include Aeolus instrument errors and forward modeling error and representativeness errors of the FV3GFS background, at the specific 25 km horizontal resolution." The error estimates show similar vertical distributions to the L2B error estimates (Garrett et al., 2022). In the future, we plan to explore the benefit of the scene-dependent L2B estimated errors on Aeolus wind assimilation. The information has been added to the text.*

Lines 192-195 : The HL method also assumes that the innovations are unbiased. How do you proceed to remove the biases in the innovations before applying the HL method to calculate the error variance ratio, which is then used in the TLS regression for estimating the wind-dependent biases?

*As now stated in the manuscript "Since the calculated innovation covariances are globally averaged over all HLOS winds, it is not surprising that the corresponding biases are small. The small residual biases in the innovations may introduce small (< 0.1) spurious spatial correlations. This spurious correlation, taken as the value calculated for the last bin (at 990 km),*

*is removed from the correlation curves at all separation distances" before calculating the HL error estimation.*

Section 2.3 : More details on the HL method are needed in this section if this manuscript is not combined with the companion article by Garrett et al. (2021).

*More details of the Hollingworth-Lonnberg method have been added to Section 2.3.*

Section 3 : This section describes the variation of the bias estimations with orbit phase, latitude, height and wind speed. However, for each orbit phase, the TLS regression is applied only as a function of vertical layer (Section 3.1) and only as a function of latitude band (Section 3.2). On the other hand, the TLS regression is applied as a function of both latitude band and vertical layer in Section 5 to estimate the bias correction values to be used in the data assimilation experiments. Since TLS regressions are not applied the same way in Sections 3 and 5, how can the results shown in Section 3 be representative of the bias estimates obtained in Section 5?  In addition, since the variance ratio vary with height, it is not clear how the TLS regression is calculated in Section 3.2.

*In section 3, we examined the height and latitude variations of TLS bias estimates for all available Aeolus winds globally and at all possible speed ranges. At the end of this section we now conclude that "The large variations of the TLS bias estimates with latitude and height guide the design of the proposed TLS bias correction in Section 5."*

*A new Fig. 13 is now added to show the TLS regressions for the latitudinal bands centered at the Equator and 80°S (where the biases are the largest according to Fig. 9). Figure 13 shows that the TLS bias estimates show considerable variations in both speed and latitude, often exceeding 1.5 m/s in magnitude at higher speeds. The text has been revised to be clearer.*

*As is now stated in Section 3.2, the vertical average of the error ratio $\delta$ is used in the calculation of the TLS bias estimates.*

Section 4 and Figure 12 : The differences between the TLS and the various OLS regressions are large, although the same linear model is used. Do you have an explanation for that? Moreover,

why the biases are underestimated (overestimated) when using FV3GFS background (Aeolus) HLOS winds as predictors?

*Although the same linear bias model is used in TLS and OLS regressions, the fitting of the regression line to Aeolus and FV3GFS winds is different. As we now conclude in Section 4 "The large differences in the bias estimates using the TLS and OLS regression are due to the fact that both Aeolus and FV3GFS winds have large errors. The fact that the errors of Aeolus winds are larger than FV3GFS background winds leads to the different weightings of Aeolus winds and FV3GFS winds in the TLS analysis (Eq. 3)."*

Section 5 : In the case this manuscript is not merged with the article by Garrett et al. (2021), I suggest adding a paragraph describing the impact of the proposed bias correction scheme on forecasts. This is important aspect to present since the main reason to develop a bias correction scheme is to remove the detrimental effect of biases on forecasts.

*Thanks for the suggestion. We have added Section 6. Here, the manuscript now summarizes the positive impact of the TLS bias correction on forecasts during the summer of 2019 that was reported by Garrett et al. In addition, Section 6 describes an Aeolus data impact experiment and forecast results for a record-breaking winter storm over the US.*

Line 360 : Publication year : 2000

*Corrected.*

Figure 9 : There are spikes in the data counts around zero-wind speed for both Mie (a) and Rayleigh (b) winds. What is the cause of this significant increase in the number of observations near this wind speed?

*Thank you for pointing this out. The spikes in the Aeolus counts around zero-wind speed were due to a bug in the plotting code. This has been fixed.*

Figures 4 and 14 : The contour interval (0.5 m/s) used in these figures is coarser than the contour interval (0.3 m/s) used in the other figures (8, 11, 12, 13). I suggest using the same 0.3 m/s

interval in Figures 4 and 14. This will improve the clarity of Figure 4 and make easier the comparison of the various panels.

*These figures are revised as suggested.*

Figure 12 : For a better assessment of the quality of the fit, I suggest adding panels showing the average O-B for both Mie and Rayleigh winds. These results are expected to be close to the average bias estimates from the TLS regression. This will prove that the best results are indeed obtained by this regression method.

*The Mie and Rayleigh innovations as a function of Aeolus winds are shown in Figure S1 below. Note that taking a simple average of the innovations results in much larger values than both the TLS regressions (treated as "truth" here) and the OLS regression of the innovations on the Aeolus winds. This seems consistent with the fact that the large random errors in Aeolus and FV3GFS winds have considerable impact on the accuracy of the OLS bias estimation and the innovations average. A new Figure 5 has been added to show the innovation scatter plots and averages as a function of Aeolus winds, together with the TLS and OLS estimates.*

*(Throughout the paper, the TLS bias estimates are assumed as the statistical "best" estimate or "truth" when compared to the OLS bias estimates. The text has been revised to be clear about this point.)*

[Figure]

*Figure S1: Vertical distributions of average innovations (top, a, b) and OLS estimated biases (bottom, c, d) for Mie (left, a, c) and Rayleigh (right, b, d) winds as a function of Aeolus winds (m/s) in descending orbits for all latitudes for the period of 1-7 September 2019. Color scale is in units of m/s.*

Figure 14 : What is the minimum number of samples used for the TLS regression? The remaining mean differences for the Mie winds above 10 km and south of 60S (Fig. 14c) look rather noisy and the remaining pattern does not look similar to the corresponding one in Figure 4. What is the reason for this?

*The minimum number of Aeolus winds used in this figure was intentionally set to 30 per 10 degree-width latitude bin in each layer to provide statistics for as many bins as possible. In the Aeolus assimilation experiments, the TLS bias correction was applied only in the bins with Aeolus wind samples > 100. This figure has been revised to reflect the latter case.*

**Response to comments of Anonymous Referee #2 (RC2) on AMT preprint "A Statistically Optimal Analysis of Systematic Differences between Aeolus HLOS Winds and NOAA's Global Forecast System" by Hui Liu et al., Atmos. Meas. Tech. Discuss., https://doi.org/10.5194/amt-2022-20. (Citation:** https://doi.org/10.5194/amt-2022-20-RC2**.)**

The paper describes a bias correction method, based on the on observation minus background statistics and relative error variances between the model background and Aeolus wind data. The paper showed that the first reprocessed HLOS winds (B10) used in this study still have large remaining mean differences compared to the FV3GFS model background. These biases vary with latitude, height, orbit phase and HLOS wind speed. To correct these biases a total least square (TLS) regression method was used. Here the authors assumed a linear dependency of the bias with wind speed. The TLS method was calculate for each orbital phase, vertical layer and 10-degree latitude band over a period of 7days. Separated for both, Mie and Raleigh channels. The necessary error variances were estimated using the Hollingsworth-Lonnberg method, which is based on obs minus background statistics. In a comparison to the normal linear regression methods the strength of the TLS method was shown.

In the following, I have only some small suggestions for some minor revisions

Lines 62-70. First, you said that NWP models have large uncertainties in regions where conventional observations are missing. Do you mean analysis uncertainties or forecast uncertainties? Also with observations, the models tend to evolve towards their own climatology over long forecast lead times because the forecast errors become much larger than the analysis errors. Can you clarify these sentences a little, and can you taken into account the different use of satellite radiances between the different centers which can explain a lot of the differences you mentioned in data sparse regions.

*Here we look at the uncertainty in the 6-h forecast. The difference shown in Fig. 1 is the zonal wind differences in 6h forecasts from the analysis times (00Z, 06Z, 12Z, and 18Z). We agree that potential different use of satellite radiances between the different NWP centers in the data sparse regions might contribute to the wind differences. The text has been revised accordingly as well as in response to reviewer #1.*

It is also not clear to me which background you use in Fig. 1 and further. In a 4d-VAR or

4D-ENVAR the background is a time series of forecasts over the used assimilation window. Which forecast lead time do you use ?

*The difference shown in Fig. 1 is the zonal wind differences in 6h forecasts from the analysis times (00Z, 06Z, 12Z, and 18Z).*

Line 78-81: I think that the use of ECMWF model fields to correct the M1 temperature bias can be neglected. Also to derive other observations like AMVs different model fields are used but their impact can be neglected almost all the time.

*The statements have been removed.*

Line 96: Innovation of Aeolus minus model is a double meaning since innovation means per definition observation minus background. Let the "Innovation of" away.

*Corrected.*

Line 99: You certainly mean Section 2 instead of section 0

*Corrected.*

Line 119-124: Here you describe the quality control steps recommended by ESA and ECMWF. Do you use additional quality control steps like background check or variational quality control check etc. ?

*The manuscript now states that "Further, a standard outlier check rejects any Aeolus wind for which $|y_i^o - y_i^b|$ is greater than 4 times the estimated errors for Aeolus winds prescribed by the data assimilation system".*

Line 166 – 182: You assume that the Aeolus bias depends linearly on the Aeolus wind speed. Do you have any prove of this assumption. The bias can also have a non-linear part and if so do you have any idea how large the non-linear part can be.

*The scatterplot and averages of Aeolus innovations as a function of Aeolus wind speed are shown in the new Figure 5. This figure suggests the existence of both linear and non-linear bias components as a function of Aeolus wind speed. In this study, we focus on the estimation of the linear component of the biases. We plan to address the nonlinear component of the biases using*

*a nonlinear TLS bias correction in a future study. This discussion has been added to the text at the end of Section 2.1.*

Chapter 3: To show, that the innovations depend on Aeolus wind speed you can include a picture where on the y-axes you display the innovations and on the x-axes you display the Aeolus wind speed.

*A new Figure 5 has been added to show this.*

Chapter 3.1: In Fig. 9 and including text you describe the success of using the TLS method to reduce the speed dependent bias. Can you explain the high peak of your data count at zero wind speed? It seems the you bias correction enhance the negative bias for large positive wind speeds in case of Rayleigh winds. Can you explain that.

*The peaks of data count at zero wind speed were due to a bug in the plotting script. It is now fixed.*

*The remaining large negative bias in the large positive speed bins of Rayleigh winds was due to a small number of samples (~50). A larger minimum number of samples (1000) of Rayleigh winds in the speed bins (sample size now ranges from ~ 1000 to 25000) is now applied and the bias is much smaller.*

Chapter 3,2; In Figure 11 there are also larger biases visible in the Mie descending orbit for small wind speeds in the tropics. Can you explain that ?

*This remaining large bias was due to a small number of Mie samples (~20) in the latitude/speed bins. A larger minimum number of samples (200) of Mie winds in these bins (sample size now ranges from ~200 to 6000) is now applied and the bias is much smaller.*

Section 4: In Fig. 12 we see that the differences between the TLS method and the different OLS regressions are relatively large although the same linear model and assumptions are used. Can you explain this ? You said that using OLS statistics underestimates the biases. How can I see that? What is the truth ?

*Although the same linear bias model is used in TLS and OLS regressions, the assumptions about the error characteristics and thus the fitting of the regression line to Aeolus and FV3GFS*

*winds is different. The TLS bias estimate is statistically optimal and it is treated as the best estimate or "truth" in this study. OLS regression of innovations as a function of either Aeolus or FV3GFS winds may have large errors due to neglecting the fact that there are errors in both winds. The OLS bias estimates are compared to the "truth" by TLS regression in Fig. 8. The text is revised to be clear about the comparison through the paper.*

At the end I think it would be a nice thing to show some assimilation and forecast results using your TLC method in comparison to not using your bias correction. At least some results from your assimilation cycle.

*The positive impact of the TLS bias correction on the forecast of a 2019 record-breaking winter storm over the US has been added in the new Section 6.*

---

## Author Response (AR2)

Review #1's comments:

I thank the authors for addressing most of my comments and accounting for my suggestions in the revised manuscript. Although the manuscript has been substantially improved, you will find below a few additional comments and corrections:

*The authors thank the reviewer for the very thorough, very careful, and very helpful suggestions and comments.*

Line 51 : The reference Liu et al. 2022 is this manuscript under review! This self-reference should be removed.

*The reference Liu et al. 2022 is removed.*

Line 186 : In the response to my comment on Line 190 (first submission), the authors state that "In the future, we plan to explore the benefit of the scene-dependent L2B estimated errors on Aeolus wind assimilation. The information has been added to the text". I however could not find this information in the revised manuscript. Could you indicate where this information had been added in the text?

*This sentence was missed and is added at the end of Section 2.3.*

Lines 281-282 : This explanation is too broad. Could you elaborate more on the role of the large random error ratio for the Rayleigh winds (Fig. 6) in the OLS regressions?

*The following is added to the text to explain the differences in the OLS and TLS bias estimates:*

*"If the predictor (either Aeolus or FV3GFS winds) has very small errors. the OLS regressions would be close to perfect. and the OLS and TLS regressions would give very similar results. In such situation, the random error ratio would be either infinity small (<< 1) or infinity large (>> 1), However, the Aeolus and FV3GFS winds have considerable errors, and the actual random error ratio is about 2-3 for the Rayleigh winds versus FV3GFS winds and about 1.2-1.5 for the Mie winds versus FV3GFS winds (Fig. 6). This leads to the large differences in the OLS and TLS bias estimates. Specifically, the OLS bias estimates using Aeolus winds as a predictor have larger differences from the TLS estimates than the OLS estimates using FV3GFS winds as a predictor."*

Fig. 14 : Panels a, b and c are nearly identical to those in Fig. 13 of the original manuscript, except panel d, which is significantly different. What is the reason for this?

*There was a bug in the plotting script for Fig. 13d of the original manuscript. This is fixed in the revised manuscript.*

Lines 342-349 : It is interesting to examine the impact of the additional bias corrections made to the Aeolus data on a particular case, such as the one presented in the new section 6. However, the fields and scores shown in Figs. 17-19 for this case study need some clarifications and further explanations. Why did you average the 0000 UTC wind vector and IVT fields over 2 days instead of showing these fields for a particular date (e.g. 0000 UTC 27 November 2019)? Why did you not average the 24-h

accumulations shown in Fig. 18 over the same forecasts as in Figs. 17 and 19 (i.e. 1200 UTC instead of 0000 UTC)?

*The wind vector and IVT fields are now shown for 0000 UTC 27 and 28, November 2019, separately. In general, similar results are found for the particular dates, as from the previous averaged statistics.*

*The caption of Figure 19 was not accurate. Now it is corrected and reads as "The forecast skill scores for 24-h accumulated precipitation for 156-h to 180 -h forecasts verified against the NCEP precipitation raingauge data analyses and validated from 1200 UTC 26 to 28 November 2019." Therefore, the statistics in Figures 18 and 19 are consistent.*

Figs. 17 and 18 : These figures show day-7 forecasts for the BASE, AEOM, AEOT experiments. I suggest adding the verifying analyses, which could be useful for assessing which experiment gives the best IVT, wind vector and precipitation accumulation forecasts.

*The ECMWF analyses of IVT and wind vector are added to the new Figures 17, and the NCEP precipitation raingauge data analyses is added to the new Figure 18. In general, the IVT, wind vector, and precipitation fields in the AEOT experiment are closer to the analyses, as demonstrated quantitively in the original Figure 19.*

Fig. 19 : Panels c and d show the differences in ETS and bias with respect of BASE and their significant levels. These panels are not discussed in the main text. Consequently, I suggest removing these plots.

*We believe that the statistical significance from panels c and d would be helpful and so we prefer to keep these panels. A reference of the statistical significance of the precipitation location forecast is added to the text.*

Line 361 : FT3GFS should be FV3GFS

*Corrected.*